# GrEASE: Generalizable Spectral Embedding with an Application to UMAP

## Abstract

Spectral Embedding (SE) is a popular method for dimensionality reduction, applicable across diverse domains. Nevertheless, its current implementations face three prominent drawbacks which curtail its broader applicability: generalizability (i.e., out-of-sample extension), scalability, and eigenvectors separation. In this paper, we introduce *GrEASE*: Generalizable and Efficient Approximate Spectral Embedding, a novel deep-learning approach designed to address these limitations. GrEASE incorporates an efficient post-processing step to achieve eigenvectors separation, while ensuring both generalizability and scalability, allowing for the computation of the Laplacian's eigenvectors on unseen data. This method expands the applicability of SE to a wider range of tasks and can enhance its performance in existing applications. We empirically demonstrate GrEASE's ability to consistently approximate and generalize SE, while ensuring scalability. Additionally, we show how GrEASE can be leveraged to enhance existing methods. Specifically, we focus on UMAP, a leading visualization technique, and introduce *NUMAP*, a generalizable version of UMAP powered by GrEASE. Our code will be publicly available upon acceptance.

## 1 Introduction

Spectral Embedding (SE) is a popular non-linear dimensionality reduction method (Belkin & Niyogi, 2003; Coifman & Lafon, 2006b), finding extensive utilization across diverse domains in recent literature. Notable applications include UMAP (McInnes et al., 2018) (the current state-of-the-art visualization method), Graph Neural Networks (GNNs) (Zhang et al., 2021; Beaini et al., 2021) and Graph Convolutional Neural Networks (GCNs) (Defferrard et al., 2016), positional encoding for Graph Transformers (Dwivedi & Bresson, 2020; Kreuzer et al., 2021) and analysis of proteins (Campbell et al., 2015; Kundu et al., 2004; Shepherd et al., 2007; Zhu & Schlick, 2021). The core of SE involves a projection of the samples into the space spanned by the leading eigenvectors of the Laplacian matrix (i.e., those corresponding to the smallest eigenvalues), derived from the pairwise similarities between the samples. SE is an expressive method which is able to preserve the global structure of high-dimensional input data, underpinned by robust mathematical foundations (Belkin & Niyogi, 2003; Katz et al., 2019; Lederman & Talmon, 2018; Ortega et al., 2018).

Despite the popularity and significance of SE, current implementations suffer from three main drawbacks: (1) *Generalizability* - the ability to directly embed a new set of test points after completing the computation on a training set (i.e., out-of-sample extension); (2) *Scalability* - the ability to handle a large number of samples within a reasonable timeframe; (3) *Eigenvectors separation* - the ability to output the *basis* of the leading eigenvectors $(v_2, \ldots, v_{k+1})$, rather than only the space spanned by them. These three properties are crucial for modern applications of SE in machine learning. Notably, the last property has attracted considerable attention in recent years (Pfau et al., 2018; Gemp et al., 2020; Deng et al., 2022; Lim et al., 2022). While most SE implementations address two of these three limitations, they often fall short in addressing the remaining one (see Sec. 2).

This paper extends the work by Shaham et al. (2018), known as SpectralNet. SpectralNet tackles the scalability and generalizability limitations of Spectral Clustering (SC), a key application of SE. However, due to a rotation and reflection ambiguity in its loss function, SpectralNet cannot directly be adapted for SE (i.e., it cannot separate the eigenvectors). In this paper, we first present a post-processing procedure to resolve the eigenvectors separation issue in SpectralNet, thereby, cre-

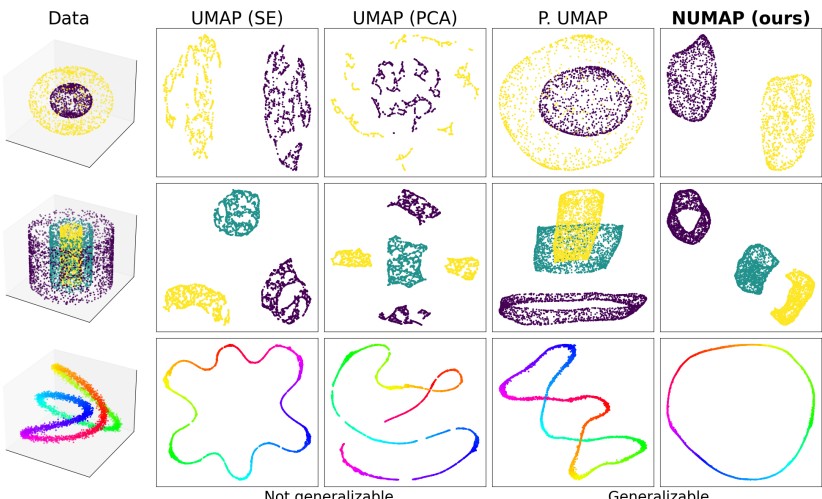

Figure 1: A comparison between non-parametric UMAP (with SE or PCA initialization), P. UMAP and NUMAP on three non-linear yet simple 3-dimensional toy datasets. NUMAP global structure abilities over P. UMAP are evident.

ating a scalable and generalizable implementation of SE, which we call *GrEASE*: Generalizable and Efficient Approximate Spectral Embedding.

GrEASE's ability to separate the eigenvectors, while maintaining generalizability and scalability offers a pathway to enhance numerous existing applications of SE and provides a foundation for developing new applications. A notable example is UMAP (McInnes et al., 2018), the current state-of-the-art visualization method. Recent work proposed Parametric UMAP (P. UMAP) (Sainburg et al., 2021) to address the UMAP lack of generalizability. However, UMAP's global structure preservation and consistency largely stem from the use of SE for initialization (Kobak & Linderman, 2021), a step absent in P. UMAP. Consequently, P. UMAP lacks a crucial component to fully replicate the performance of UMAP, especially in terms of global structure preservation. Nonetheless, a series of studies have incorporated P. UMAP, underscoring the significant impact of a generalizable version of UMAP (Xu & Zhang, 2023; Eckelt et al., 2023; Leon-Medina et al., 2021; Xie et al., 2023; Yoo et al., 2022).

In this paper, we also introduce a novel application of GrEASE for generalizable UMAP, which we term *NUMAP*. NUMAP integrates the UMAP loss with SE initialization, similar to the original non-parametric UMAP. As a result, NUMAP achieves comparable results to UMAP, while also offering generalization capabilities. Fig. 1 depicts this idea. P. UMAP fails to preserve the global structure (e.g., separation of the clusters), while UMAP and NUMAP does so successfully.

Our contributions can be summarized as follows: (1) We introduce GrEASE, a novel approach for generalizable approximate SE; (2) We establish a foundation for a range of new SE applications and enhancements to existing methods; (3) We present NUMAP: a novel application of GrEASE for generalizable UMAP; (4) We propose a new evaluation method for dimensionality reduction methods, which enables quantification of global structure preservation.

## 2 RELATED WORK

Current SE implementations typically address two out of its three primary limitations: generalizability, scalability, and eigenvector separation. Below, we outline key implementations that tackle each pair of these challenges. Following this, we discuss recent works related to eigenvectors separation and generalizable visualizations techniques.

**Scalable with eigenvectors separation.** Popular implementations of SE are mostly based on sparse matrix decomposition techniques (e.g., ARPACK (Lehoucq et al., 1998), AMG (Brandt et al.,

1984), LOBPCG (Benner & Mach, 2011)). These methods are relatively scalable, as they are almost linear in the number of samples. Nevertheless, their out-of-sample extension is far from trivial. Usually, it is done by out-of-sample extension (OOSE) methods such as Nyström extension method (Nyström, 1930) or Geometric Harmonics (Coifman & Lafon, 2006a; Lafon et al., 2006). However, these methods provide only local extension (i.e., near existing training points), and are both computationally and memory restrictive, as they rely on computing the distances between every new test point and all training points.

**Scalable and generalizable.** Several approaches to SC approximate the space spanned by the first eigenvectors of the Laplacian matrix, which is sufficient for clustering purposes, and can also benefit other specific applications. For example, SpectralNet (Shaham et al., 2018) leverages deep neural networks to approximate the first eigenfunctions of the Laplace-Beltrami operator in a scalable manner, thus also enabling fast inference of new unseen samples. BASiS (Streicher et al., 2023) achieves these goals using affine registration techniques to align batches. However, these methods' inability to separate the eigenvectors prevents their use in many modern applications.

**Generalizable with eigenvectors separation.** Another proposed approach to SE is Diffusion-Net (Mishne et al., 2019), a deep-learning framework for generalizable Diffusion Maps embedding (Coifman & Lafon, 2006b), which is similar to SE. However, the training procedure of the network is computationally expensive, therefore restricting its usage for large datasets.

In contrast, we introduce GrEASE, which generalizes the separated eigenvectors to unseen points with a single feed-forward operation, while maintaining scalability.

**Eigenvectors separation.** Extensive research has been conducted on the eigenvectors separation problem, both within and beyond the spectral domain (Lim et al., 2022; Ma et al., 2024). However, recent suggestions are constrained computationally, both by extensive run-time and memory consumption. For example, Pfau et al. (2018) proposed a solution to this issue by masking the gradient information from the loss function. However, this approach necessitates the computation of full Jacobians at each time step, which is highly computationally intensive. Gemp et al. (2020) employs an iterative method to learn each eigenvector sequentially. Namely, they learn an eigenvector while keeping the others frozen. This process has to be repeated $k$ times (where $k$ is the embedding dimension), which makes this approach also computationally expensive. Deng et al. (2022) proposed an improvement of the latter, by parallel training of $k$ NNs. However, as discussed in their paper, this approach becomes costly for large values of $k$. Furthermore, it necessitates retaining $k$ trained networks in memory, which leads to significant memory consumption. Chen et al. (2022) proposed a post-processing solution to this problem using the Rayleigh-Ritz method. However, this approach involves the storage and multiplication of very large dense matrices, rendering it impractical for large datasets. In contrast, GrEASE offers an efficient one-shot post-processing solution to the eigenvectors separation problem.

**Generalizable visualization.** Several works have attempted to develop parametric approximations for non-parametric visualization methods, in addition to Parametric UMAP (P. UMAP) (Sainburg et al., 2021). Notable examples include (Van Der Maaten, 2009) and (Kawase et al., 2022), which use NNs to make t-SNE generalizable, and (Schofield & Lensen, 2021), which aims to make UMAP more interpretable. However, P. UMAP has demonstrated superior performance. NUMAP presents a method to surpass P. UMAP in terms of global structure preservation.

## 3 PRELIMINARIES

In this section, we begin by providing the fundamental definitions that will be used throughout this work. Additionally, we briefly outline the key components of UMAP and P. UMAP.

### 3.1 SPECTRAL EMBEDDING

Let $\mathcal{X} = \{x_1, \dots, x_n\} \subseteq \mathbb{R}^d$ denote a collection of unlabeled data points drawn from some unknown distribution $\mathcal{D}$. Let $W \in \mathbb{R}^{n \times n}$ be a positive symmetric graph affinity matrix, with nodes corresponding to $\mathcal{X}$, and let $D$ be the corresponding diagonal degree matrix (i.e. $D_{ii} = \sum_{j=1}^{n} W_{ij}$).

The Unnormalized Graph Laplacian is defined as $L = D - W$. Other normalized Laplacian versions are the Symmetric Laplacian $L_{\text{sym}} = D^{-\frac{1}{2}} L D^{-\frac{1}{2}}$ and the Random-Walk (RW) Laplacian $L_{\text{rw}} = D^{-1} L$. GrEASE is applicable to all of these Laplacian versions. The eigenvalues of $L$ can be sorted to satisfy $0 = \lambda_1 \leq \lambda_2 \leq \cdots \leq \lambda_n$ with corresponding eigenvectors $v_1, \ldots, v_n$ (Von Luxburg, 2007). It is important to note that the first pair (i.e., $\lambda_1, v_1$) is trivial - for every Laplacian matrix $\lambda_1 = 0$, and for the unnormalized and RW Laplacians $v_1 = \frac{1}{\sqrt{n}} \vec{1}$, namely the constant vector.

For a given target dimension $k$, the first non-trivial $k$ eigenvectors provide a natural non-linear low-dimensional embedding of the graph which is known as *Spectral Embedding* (SE). In practice, we denote by $V \in \mathbb{R}^{n \times k}$ the matrix containing the first non-trivial $k$ eigenvectors of the Laplacian matrix as its columns (i.e., $v_2, \ldots, v_{k+1}$). The SE representation of each sample $x_i \in \mathbb{R}^d$ is the $i$th row of $V$, i.e., $y_i = (v_2(i), \ldots, v_{k+1}(i))$.

## 3.2 SPECTRALNET

A prominent method for addressing scalability and generalizability in Spectral Clustering (SC) is using deep neural networks, for example SpectralNet (Shaham et al., 2018). SpectralNet follows a common methodology for transferring the problem of matrix decomposition to its smallest eigenvectors to an optimization problem, through minimization of the Rayleigh Quotient (RQ).

**Definition 1.** *The Rayleigh quotient (RQ) of a Laplacian matrix $L \in \mathbb{R}^{n \times n}$ is a function $R_L : \mathbb{R}^{n \times k} \to \mathbb{R}$ defined by*

$$R_L(A) = \text{Tr}(A^T L A)$$

SpectralNet first minimizes the RQ on small batches, while enforcing orthogonality. Namely, it approximates $\theta^* = \arg\min_\theta \frac{1}{m^2} R_L(f_\theta(X))$ $s.t.$ $\frac{1}{m} f_\theta(X)^T f_\theta(X) = I_{k \times k}$. Thereby, it learns a map $f : \mathbb{R}^d \to \mathbb{R}^k$ (where $d$ is the input dimension) which approximates the space spanned by the first $k$ eigenfunctions of the Laplace-Beltrami operator on the underlying manifold $\mathcal{D}$ (Belkin & Niyogi, 2006; Shi, 2015). Following this, it clusters the representations via KMeans. These eigenfunctions are a natural generalization of the SE to unseen points, enabling both scalable and generalizable spectral clustering.

## 3.3 UMAP AND PARAMETRIC UMAP

UMAP (McInnes et al., 2018) is the current state-of-the-art visualization method. UMAP presents a significant advancement over previous methods, primarily due to its enhanced scalability and superior ability to preserve global structure. This approach involves the construction of a graph from the input high-dimensional data and the learning of a low-dimensional representation. The objective is to minimize the KL-divergence between the input data graph and the representation graph.

However, as discussed in (Kobak & Linderman, 2021), UMAP primarily derives its global preservation abilities, as well as its consistency, from initializing the representations using SE. Therefore, the SE initialization serves as a critical step for UMAP to uphold the global structure (see demonstration in Fig. 1). Global preservation, in this context, refers to the separation of different classes, and avoiding the separation of existing classes. We refer the reader to (Kobak & Linderman, 2021) for a more comprehensive discussion about the effects of informative initialization on UMAP's performance.

UMAP method can be divided into three components (summarized in Fig. 3): (1) constructing a graph which best captures the global structure of the input data; (2) initializing the representations via SE; (3) Learning the representations, via SGD, which best capture the original graph. This setup does not facilitate generalization, as both steps (2) and (3) lack generalizability.

Recently, a generalizable version of UMAP, known as Parametric UMAP (P. UMAP), was introduced (Sainburg et al., 2021). P. UMAP replaces step (3) with the training of a neural network. Importantly, it overlooks step (2), the SE initialization. Consequently, P. UMAP struggles to preserve global structure, particularly when dealing with non-linear structures. Fig. 1 illustrates this phenomenon with several non-linear yet simple structures. Noticeably, P. UMAP fails to preserve global structure (e.g., it does not separate different clusters).

## 4 METHOD

### 4.1 MOTIVATION

It is well known that the matrix $V$, containing the first $k$ eigenvectors of $L$ (i.e., those corresponding to the $k$ smallest eigenvalues) as its columns, minimizes $R_L(A)$ under orthogonality constraint (i.e. $A^T A = I$) (Li, 2015).

However, a rotation and reflection ambiguity of the RQ prohibits a trivial adaptation of this concept to SE. Basic properties of trace imply that for any orthogonal matrix $Q \in \mathbb{R}^{k \times k}$ the matrix $U := VQ$ satisfies

$$R_L(U) = \text{Tr}(U^T L U) = \text{Tr}(Q^T V^T L V Q) = \text{Tr}(Q Q^T V^T L V) = \text{Tr}(V^T L V) = R_L(V)$$

Thus, every such $U$ also minimizes $R_L$ under the orthogonality constraint, and therefore this kind of minimization solely is missing eigenvectors separation, which is crucial for many applications.

### 4.2 GrEASE

In fact, we prove that the aforementioned form $VQ$ is the only form of a minimizer of $R_L$ under the orthogonality constraint.

**Lemma 1.** *Every minimizer of $R_L$ under the orthogonality constraint, is of the form $VQ$, where $V$ is the first $k$ eigenvectors matrix of $L$ and $Q$ is an arbitrary squared orthogonal matrix.*

The proof of Lemma 1 appears in Appendix A.

Lemma 1 implies that SpectralNet's method, using a deep neural network for RQ minimization (while enforcing orthogonality), does not lead to the SE. However, it only leads to the space spanned by the constant vector and the leading $k-1$ eigenvectors of $L$, with different rotations and reflections for each run. Therefore, each time the RQ is minimized, it results in a different linear combination of the smallest eigenvectors. Although this is sufficient for clustering purposes, as we search for reproducibility, consistency, and separation of the eigenvectors, the RQ cannot solely provide the SE, necessitating the development of new techniques in GrEASE.

**Setup.** Here we present the two key components of GrEASE, a scalable and generalizable SE method. We consider the following setup: Given a training set $\mathcal{X} \subseteq \mathbb{R}^d$ and a target dimension $k$, we construct an affinity matrix $W$, and compute an approximation of the leading eigenvectors of its corresponding Laplacian. In practice, we first utilize SpectralNet (Shaham et al., 2018) to approximate the space spanned by the first $k + 1$ eigenfunctions of the corresponding Laplace-Beltrami operator, and then find each of the $k$ leading eigenfunctions within this space (i.e. the SE). Namely, GrEASE computes a map $F_\theta : \mathbb{R}^d \to \mathbb{R}^k$, which approximates the map $\bar{f} = (f_2, \ldots, f_{k+1})$, where $f_i$ is the $i$th eigenfunction of the Laplace-Beltrami operator on the underlying manifold $\mathcal{D}$.

**Eigenspace approximation.** As empirically showed in (Shaham et al., 2018), and motivated from Lemma 1, SpectralNet loss is minimized when $F_\theta = T \circ (f_1, \ldots, f_{k+1})$, where $T : \mathbb{R}^{k+1} \to \mathbb{R}^{k+1}$ is an arbitrary isometry. Namely, $F_\theta$ approximates the space spanned by the first $k + 1$ eigenfunctions. However, the SE (i.e. each of the leading eigenfunctions) is poorly approximated. Each time the RQ is minimized, the eigenfunctions are approximated up to a different isometry $T$. Fig. 2a demonstrates this phenomenon on the toy moon dataset - a noisy half circle linearly embedded into 10-dimension input space (see Sec. 5.1). Employing SpectralNet approach indeed enables us to consistently achieve a perfect approximation of the space (i.e., the errors are accumulated around 0). However, when comparing vector to vector, it becomes apparent that the SE was seldom attained.

**SE approximation.** To get the SE consistently (i.e., to separate the eigenvectors), we suggest a simple use of Lemma 1. Notice that based on Lemma 1 we can compute a rotated version of the diagonal eigenvalues matrix. Namely,

$$(VQ)^T L (VQ) = Q^T V^T L V Q = Q^T \Lambda Q =: \tilde{\Lambda}$$

Where $\Lambda$ is the diagonal eigenvalues matrix. Due to the uniqueness of eigendecomposition, the eigenvectors and eigenvalues of the small matrix $\tilde{\Lambda} \in \mathbb{R}^{k+1 \times k+1}$ are $Q^T$ and $diag(\Lambda)$, respectively.

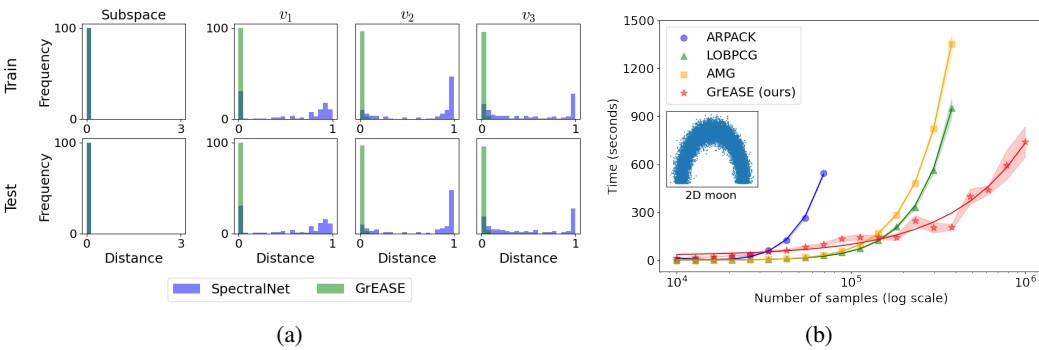

Figure 2: (a) Approximation of the 2-dimensional SE of the moon dataset using SpectralNet (in blue) and GrEASE (in green) over 100 runs, on train (top row) and test (bottom row). Left column: distribution of the Grassmann distance between the output and true subspace. Second to Fourth columns: distribution of the $sin^2$ distance between each output and true eigenvector separately. Evidently, GrEASE is able to separate the eigenvectors. (b) Running times of SE using GrEASE vs. other methods on the Moon dataset (a 2D moon linearly embedded into 10D input space), relative to the number of samples, and with standard deviation confidence intervals. Evidently, GrEASE is the fastest asymptotically.

Hence, by diagonalizing $\tilde{\Lambda}$ we get the eigenvalues and are also able to separate the eigenvectors (i.e., approximate the SE).

In practice, as $Q$ is a property of SpectralNet optimization (manifested by the parameters), we compute the matrix $\tilde{\Lambda}$ by averaging over a few random minibatches and diagonalize it. Thereby, making this addition very cheap computationally. The eigenvectors matrix of $\tilde{\Lambda}$ is the inverse of the orthogonal matrix $Q$, and hence by multiplying the output of the learned map $F_\theta$ by this matrix, the SE is retained. Also, the eigenvalues of $\tilde{\Lambda}$ are the eigenvalues of $L$.

The effect of this intentional rotation is represented in the Fig. 2a. GrEASE was not only able to consistently approximate the space, but also approximate each eigenvector. While SpectralNet errors are distributed over a large range of values, GrEASE errors are small, capturing only the smallest error bin in the figure.

**Algorithms Layout.** Our end-to-end training approach is summarized in Algorithms 1 and 2 in Appendix B. We run them consecutively: First, we train $F_\theta$ to approximate the first eigenfunctions up to isometry (Algorithm 1) (Shaham et al., 2018). Second, we find the matrices $Q^T$ and $\Lambda$ to separate the eigenvectors and retrieve the SE and its corresponding eigenvalues (Algorithm 2). App. C details additional considerations about the implementation.

Once we have $F_\theta$ and $Q^T$, computing the embeddings of the train set or of new test points (i.e., out-of-sample extension) is straightforward: we simply propagate each test point $x_i$ through the network $F_\theta$ to obtain their embeddings $\tilde{y}_i$, and use $Q^T$ to get the SE embeddings $y_i = \tilde{y}_i Q^T$.

**Time and Space complexity.** As the network iterates over small batches, and the post-processing operation is much cheaper, GrEASE's time complexity is approximately linear in the number of samples. This is also demonstrated in Fig. 2b, where the continuous red line, representing linear regression, aligns with our empirical results. App. C provides a discussion about the complexity of GrEASE. Note also that GrEASE is much more memory-efficient than existing methods, as it does not require storing the full graph, or any large matrix, in the memory, but rather one small graph or matrix (of a minibatch) at a time.

### 4.3 NUMAP

We focus on GrEASE application to UMAP, one of many methods which can benefit from a generalizable SE. As discussed in Sec. 3.3, the SE initialization is crucial for the global preservation

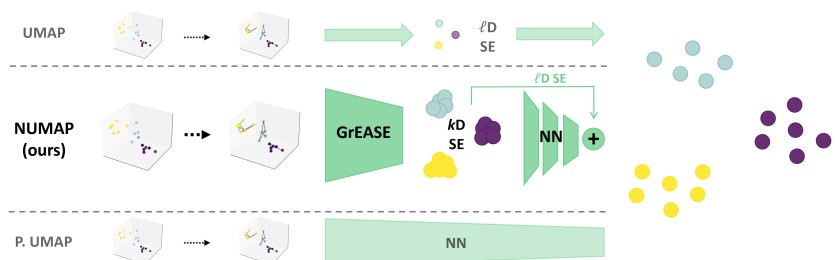

Figure 3: UMAP vs. NUMAP vs. P. UMAP overview. A green arrow represents a non-parametric step. NUMAP integrates SE, as in UMAP, while enabling generalization.

abilities of UMAP. Therefore, we seek a method to incorporate SE into a generalizable version of UMAP. It is important to note that a naive approach would be to fine-tune GrEASE using UMAP loss. However, during this implementation, we encountered the phenomenon of catastrophic forgetting (see App. F).

The core of our idea is illustrated in Fig. 3. Initially, we use GrEASE to learn a parametric representation of the $k$-dimensional SE of the input data. Subsequently, we train an NN to map from the SE to the UMAP embedding space, utilizing UMAP contrastive loss. The objective of the second NN is to identify representations that best capture the local structure of the input data graph. SE transforms complex non-linear structures into simpler linear structures, allowing the second NN to preserve both local and global structures effectively. To enhance this capability, we incorporate residual connections from the first to the last layer of the second NN. Specifically, the objective is to minimize the residual between the $\ell$-dimensional UMAP embedding and the $\ell$-dimensional SE. Note that this could not have been made possible without GrEASE's ability to separate the eigenvectors (and would not be practical without its inherent generalizability and scalability). Fig. 1 demonstrates this capability with several simple structures.

### 4.4 ADDITIONAL APPLICATIONS

In this section we seek to highlight GrEASE's potential impact on important tasks and applications (besides UMAP), as it integrates generalizability, scalability and eigenvectors separation. As discussed in Sec. 1, SE is applied across various domains, many of which can benefit generalizability capabilities by simply replacing the current SE implementation with GrEASE. We therefore elaborate herein the significance of SE in selected applications, and discuss how GrEASE, as a generalizable approximation of it, can enhance their effectiveness and applicability.

**Fiedler vector and value.** A special case of SE is the Fiedler vector and value (Fiedler, 1973; 1975). The Fiedler value, also known as algebraic connectivity, refers to the second eigenvalue of the Laplacian matrix, while the Fiedler vector refers to the associated eigenvector. This value quantifies the connectivity of a graph, increasing as the graph becomes more connected. Specifically, if a graph is not connected, its Fiedler value is 0. The Fiedler vector and value are a main topic of many works (Andersen et al., 2006; Barnard et al., 1993; Kundu et al., 2004; Shepherd et al., 2007; Cai et al., 2018; Zhu & Schlick, 2021; Tam & Dunson, 2020).

As GrEASE is able to distinguish between the eigenvectors and approximate the eigenvalues, it has the capability to approximate both the Fiedler vector and value, while also generalizing the vector to unseen samples (see Sec. 5.1).

**Diffusion Maps.** A popular method which incorporates SE, alongside the eigenvalues of the Laplacian matrix, is Diffusion Maps (Coifman & Lafon, 2006b). Diffusion Maps embeds a graph (or a manifold) into a space where the pairwise Euclidean distances are equivalent to the pairwise Diffusion distances on the graph.

In practice, for an $k$-dimensional embedding space and a given $t \in \mathbb{N}$, Diffusion Maps maps the points to the leading eigenvectors of the RW-Laplacian matrix of the data as follows:

$$X \rightarrow \left( (1 - \lambda_2)^t v_2 \quad \cdots \quad (1 - \lambda_{k+1})^t v_{k+1} \right) = Y$$

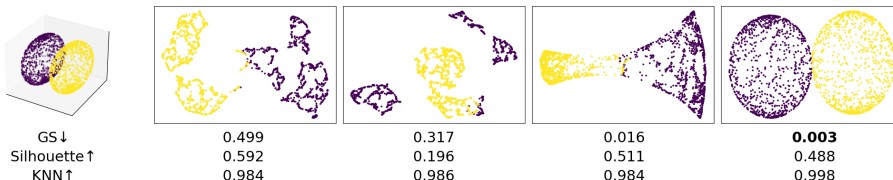

| | | | | |
|---|---|---|---|---|
| GS↓ | 0.499 | 0.317 | 0.016 | **0.003** |
| Silhouette↑ | 0.592 | 0.196 | 0.511 | 0.488 |
| KNN↑ | 0.984 | 0.986 | 0.984 | 0.998 |

Figure 4: A demonstration of the alignment between the intuitive expectation and the Grassmann Score (GS) results on a toy dataset of two 3-dimensional tangent spheres. Four possible 2-dimensional embeddings of this dataset are provided, along with their corresponding GS, KNN accuracy and Silhouette score. Unlike KNN and Silhouette, GS effectively captures the preservation of global structure.

Where $X \in \mathbb{R}^{n \times d}$ is a matrix containing each input point as a row, and $Y \in \mathbb{R}^{n \times k}$ is a matrix containing each of the representations as a row. As GrEASE is able to approximate both the eigenvectors and eigenvalues of the Laplacian matrix, it is able to make Diffusion Maps generalizable and efficient (Sec. 5.1).

### 4.5 EVALUATING UMAP EMBEDDING - GRASSMANN SCORE

Common evaluation methods for dimensionality reduction, particularly for visualization, are predominantly focused on local structures. For instance, McInnes et al. (2018); Kawase et al. (2022) use KNN accuracy and Trustworthiness, which only account for the local neighborhoods of each point while overlooking global structures such as cluster separation. One global evaluation method is the Silhouette score, which measures the clustering quality of the classes within the embedding space. However, this score does not capture the preservation of the overall global structure.

To address this gap, we propose a new evaluation method, specifically appropriate for assessing global structure preservation in graph-based dimensionality reduction methods (e.g., UMAP, t-SNE). The leading eigenvectors of the Laplacian matrix are known to encode crucial global information about the graph (Belkin & Niyogi, 2003). Thus, we measure the distance between the global structures of the original and embedding manifolds using the Grassmann distance between the first eigenvectors of their respective Laplacian matrices. We refer to this method as the *Grassmann Score* (GS).

It is important to note that GS includes a hyper-parameter - the number of eigenvectors considered. Increasing the number of eigenvectors incorporates more local structure into the evaluation. A natural choice for this hyperparameter is 2, which corresponds to comparing the Fiedler vectors (i.e., the second eigenvectors of the Laplacian). The Fiedler vector is well known for encapsulating the global information of a graph (Fiedler, 1973; 1975). Unless stated otherwise, we use two eigenvectors for computing the GS. Fig. 4 demonstrates GS (alongside Silhouette and KNN scores for comparison) on a few embeddings of two tangent spheres, independently to the embedding methods. Notably, the embedding on the right appears to best preserve the global structure, as indicated by the smallest GS value. In contrast, the KNN scores are comparable across all embeddings (e.g., KNN ignores separation of an existing class), and the Silhouette score even favors other embeddings. In App. D we mathematically formalize GS and provide additional examples of embeddings and their corresponding GS. These examples further support the intuition that GS effectively captures global structure preservation better than previous measures.

## 5 EXPERIMENTS

### 5.1 EIGENVECTORS SEPARATION - GENERALIZABLE SE

In this section, we demonstrate GrEASE's ability to approximate and generalize the SE using four real-world datasets: CIFAR10 (via their CLIP embedding); Appliances Energy Prediction dataset (Candanedo, 2017); Kuzushiji-MNIST (KMNIST) dataset (Clanuwat et al., 2018); Parkinsons Telemonitoring dataset (Tsanas & Little, 2009). Particularly, we compare our results with SpectralNet, which has been empirically shown to approximate the SE space. However, as our results demon-

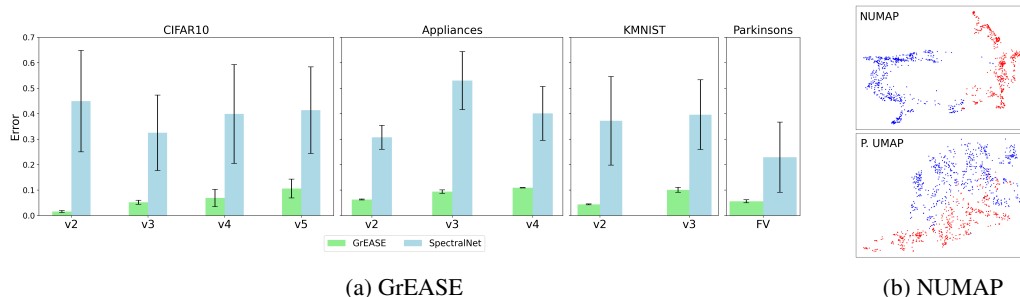

(a) GrEASE           (b) NUMAP

Figure 5: (a) A comparison between GrEASE and SpectralNet SE and Fiedler Vector (FV) approximation on real-world datasets. The values are the mean and standard deviation of the $sin^2$ distance between the predicted and true eigenvector of the test set, over 10 runs. Lower is better. GrEASE ability to separate the eigenvectors is evident. (b) Banknote's visualization by NUMAP and P. UMAP. A better separation between classes is observed in NUMAP.

strate, SpectralNet is insufficient for accurately approximating SE. For additional technical details regarding the datasets, architectures and training procedures, we refer the reader to Appendix G.

**Evaluation Metrics.** To assess the approximation of each eigenvector (i.e., the SE), we compute the $sin^2$ of the angle between the predicted and true vectors. This can be viewed as the 1-dimensional case of the Grassmann distance, a well-known metric for comparing equidimensional linear subspaces (see formalization in App. D). Concerning the eigenvalues approximation evaluation, we measure the Pearson Correlation between the predicted and true eigenvalues (computed via SVD).

Fig. 5a presents our results on the real-world datasets. GrEASE's output is used directly, while SpectralNet's predicted eigenvectors are resorted to minimize the mean $sin^2$ distance. The results clearly show that GrEASE consistently produces significantly more accurate SE approximations compared to SpectralNet, due to the improved separation of the eigenvectors.

Additionaly, note the GrEASE approximates the eigenvalues as well. When concerning a series of Laplacian eigenvalues, the most important property is the relative increase of the eigenvalues (Coifman & Lafon, 2006b). GrEASE demonstrates a strong ability to approximate this property. To see this, we repeated GrEASE's eigenvalue approximation (10 times) and calculated the Pearson correlation between the predicted and accurate eigenvalues vector. We compared the first 10 eigenvalues. The resulting mean correlation and standard deviation are: Parkinsons Telemonitoring: $\mathbf{0.917}_{\pm 0.0381}$; Appliances Energy Prediction: $\mathbf{0.839}_{\pm 0.0342}$;

## 5.2 SCALABILITY

Noteworthy, GrEASE not only generalizes effectively but also does so more quickly than the most scalable (yet non-generalizable) existing methods. Fig. 2b demonstrates this point on the toy moon dataset - a 2D moon linearly embedded into 10D input space. To evaluate scalability, we measured the computation time required for SE approximation, for an increasing number of samples. We compared the results with the three most popular methods for sparse matrix decomposition, which are currently the fastest implementations: ARPACK (Lehoucq et al., 1998), LOBPCG (Benner & Mach, 2011), and AMG (Brandt et al., 1984). For each number of samples, we calculated the Laplacian matrix that is 99% sparse. Each method was executed five times, initialized with different seeds. As discussed in Sec. 4, GrEASE demonstrates approximately linear time complexity, and indeed, for higher numbers of samples, GrEASE converges significantly faster.

## 5.3 NUMAP - GENERALIZABLE UMAP

In this section, we demonstrate NUMAP's ability to preserve global structure, while enabling fast inference of test points. We compare our results with P. UMAP. We begin with showcasing NUMAP's capacity to preserve global structure using three toy datasets. These examples are particularly in-

Table 1: A comparison between NUMAP and P. UMAP visualization on real-world datasets. The values are the mean and standard deviation of the measures on the test set, over 5 runs. NUMAP is superior in preserving global structure.

| Metric | Method | Cifar10 | Appliances | Wine | Banknote |
|--------|--------|---------|------------|------|----------|
| KNN $\uparrow$ | NUMAP | $0.764_{\pm 0.044}$ | $0.946_{\pm 0.020}$ | $0.972_{\pm 0.020}$ | $0.946_{\pm 0.056}$ |
|  | P. UMAP | $0.880_{\pm 0.007}$ | $0.992_{\pm 0.003}$ | $0.961_{\pm 0.058}$ | $0.944_{\pm 0.030}$ |
| GS $\downarrow$ | NUMAP | $\mathbf{0.158}_{\pm 0.062}$ | $\mathbf{0.554}_{\pm 0.031}$ | $\mathbf{0.437}_{\pm 0.076}$ | $\mathbf{0.626}_{\pm 0.056}$ |
|  | P. UMAP | $0.195_{\pm 0.087}$ | $0.719_{\pm 0.242}$ | $0.563_{\pm 0.104}$ | $0.686_{\pm 0.024}$ |

sightful, as P. UMAP fails to visualize correctly even these simple datasets. Following this, to further demonstrate NUMAP's effectiveness in preserving global structure, we present quantitative results on real-world datasets: CIFAR10 (via their CLIP embedding); Appliances Energy Prediction dataset; Wine (Aeberhard & Forina, 1992); Banknote Authentication (Lohweg, 2012). For additional technical details regarding the datasets, architectures and training procedures, we refer the reader to Appendix G.

**Evaluation Metrics.** To evaluate and compare the embeddings, we employed both local and global evaluation metrics. For local evaluation, we used the well-established accuracy of a KNN classifier on the embeddings (McInnes et al., 2018; Sainburg et al., 2021). For global evaluation, we use GS (see discussion in Sec. 4.5).

**Synthetic data.** Fig. 1 presents three simple non-linear synthetic 3-dimensional structures and their 2-dimensional visualizations using UMAP (non-parametric), P. UMAP and NUMAP. UMAP (using its default configuration, SE initialization) accurately preserves the global structure in its 2-dimensional representations, but lack the ability to generalize to unseen points. Among the generalizable methods (i.e., P. UMAP and NUMAP), P. UMAP fails to preserve the global structure: in the top two rows, it does not separate the clusters, while in the bottom row, it introduces undesired color overlaps. In contrast, NUMAP effectively preserves these separations and avoids the unnecessary overlapping.

**Real-world data.** Tab. 1 presents our results on real-world datasets. The local (i.e., KNN) results are comparable with P. UMAP. However, NUMAP better captures the global structure (based on the lower GS). In other words, NUMAP achieves comparable local preservation results with P. UMAP, while possessing more global structure expressivity. The synthetic datasets emphasize the importance of global structure preservation. Fig. 5b further demonstrates NUMAP's superior ability to preserve global structure, as evidenced by the improved class separation in the Banknote dataset.

## 6 CONCLUSIONS

We first introduced GrEASE, a deep-learning approach for approximate SE. GrEASE addresses the three primary drawbacks of current SE implementation: generalizability, scalability and eigenvectors separation. By incorporating a post-processing diagonalization step, GrEASE enables eigenvectors separation without compromising generalizability or scalability. Remarkably, this one-shot post-processing operation lays the groundwork for a wide range of new applications of SE, which would not have been possible without its scalable and generalizable implementation. It also presents a promising pathway to enhance current applications of SE.

In particular, we presented NUMAP, a novel application of GrEASE for generalizable UMAP visualization. We believe the integration of SE with deep learning can have a significant impact on unsupervised learning methods. Further research should delve into exploring the applications of SE across various fields.

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

# A    PROOF OF LEMMA 1

First, we remind an important property of the Rayleigh Quotient.

**Remark 1.** *The Rayleigh Quotient of a positive semi-definite matrix $L \in \mathbb{R}^{n \times n}$ with eigenvectors $v_1, \ldots, v_n$ corrisponding to the eigenvalues $\lambda_1 \leq \cdots \leq \lambda_n$, $R_L$ satisfies $arg \min_{||v||=1} R_L(v) = v_1$ and for each $i > 1$ $arg \min_{||v||=1} R_L(v) = v_i$ for $v \perp v_1, \ldots, v_{i-1}$ (Li, 2015).*

**Lemma 1.** Let $L \in \mathbb{R}^{n \times n}$ be an Unnormalized Laplacian matrix and $R_L : O(n, k) \to \mathbb{R}$ its corresponding RQ, and Let $A$ be a minimizer of $R_L$. Denote $V \in \mathbb{R}^{n \times k}$ as the matrix containing the first $k$ eigenvectors of $L$ as its columns, and $\Lambda$ the corresponding diagonal eigenvalues matrix. Then, there exists an orthogonal matrix $Q \in \mathbb{R}^{k \times k}$ such that $A = VQ$.

*Proof.* As $V$ minimizes $R_L$, we get that $min_U R_L(U) = R_L(V) = \sum_{i=1}^{k} \lambda_i$, where $0 = \lambda_1 \leq \lambda_2 \leq \cdots \leq \lambda_n$ are the eigenvalues of $L$. This yields

$$R_L(A) = \text{Tr}(A^T L A) = \sum_{i=1}^{k} \lambda_i$$

$A^T L A$ is symmetric, and hence orthogonally diagonalizable, which means there exists an orthogonal matrix $Q \in \mathbb{R}^{k \times k}$ and a diagonal matrix $D \in \mathbb{R}^{k \times k}$ s.t.

$$A^T L A = Q^T D Q$$

Which can be written as

$$(AQ^T)^T L (AQ^T) = D$$

Denoting by $d_1, \ldots, d_k$ the diagonal values of $D$, the last equation yields

$$\sum_{i=1}^{k} d_i = R_L(AQ^T) = R_L(A) = \sum_{i=1}^{k} \lambda_i$$

Note that based on Remark 1 $\lambda_i \leq d_i$ for each $i$, as $AQ^T \in O(n, k)$. Hence, $d_i = \lambda_i$, i.e.,

$$(AQ^T)^T L (AQ^T) = \Lambda$$

As the eigendecomposition of a matrix is unique, this yields $AQ^T = V$, which means $A = VQ$. $\square$

# B    ALGORITHM LAYOUTS

---

**Algorithm 1:** SpectralNet training (Shaham et al., 2018)

---

**Input:** $\mathcal{X} \subseteq \mathbb{R}^d$, number of dimensions $k$, batch size $m$
**Output:** Trained $F_\theta$ which approximates the first $k + 1$ eigenfunctions up to isometry

1 Randomly initialize the network weights $\theta$
2 **while** $\mathcal{L}(\theta)$ *not converged* **do**
3     **Orthogonalization step:**
4     Sample a random minibatch $X$ of size $m$
5     Forward propagate $X$ and compute inputs to orthogonalization layer $\tilde{Y}$
6     Compute the $QR$ factorization $QR = \tilde{Y}$
7     Set the weights of the orthogonalization layer to be $\sqrt{m} R^{-1}$
8     **Gradient step:**
9     Sample a random minibatch $x_1, \ldots, x_m$
10     Compute the $m \times m$ affinity matrix $W$
11     Forward propagate $x_1, \ldots, x_m$ to get $y_1, \ldots, y_m$
12     Compute the loss $\mathcal{L}(\theta)$(Sec. 3.2)
13     Use the gradient of $\mathcal{L}(\theta)$ to tune all $F_\theta$ weights, except those of the output layer;

---

---

**Algorithm 2:** Eigenvectors separation

**Input:** $\mathcal{X} \subseteq \mathbb{R}^d$, batch size $m$, Trained $F_\theta$ which approximates the first $k + 1$
        eigenfunctions up to isometry
**Output:** $F_\theta$ which approximates the leading eigenfunctions

1 n_iterations $\leftarrow \lfloor \frac{|\mathcal{X}|}{m} \rfloor$
2 sample n_iterations minibatches $X_i \in \mathbb{R}^{m \times d}$
3 Forward propogate all $X_i$ and obtain $F_\theta$ outputs $Y_i \in \mathbb{R}^{m \times k+1}$
4 Compute the $m \times m$ affinity matrices $W_i$
5 compute all corresponding RW-Laplacians $L_i$
6 $\tilde{\Lambda} \leftarrow \frac{1}{\text{n\_iterations}} \sum_i Y_i^T L_i Y_i$
7 Diagonalize $\tilde{\Lambda}$ to get $\tilde{Q}^T$ and the leading eigenvalues
8 Sort the leading eigenvalues, and the columns $\tilde{Q}^T$ correspondingly
9 $Q^T \leftarrow$ last $k$ columns of $\tilde{Q}^T$

---

## C  IMPLEMENTATION'S ADDITIONAL CONSIDERATIONS

### C.1  TIME AND SPACE COMPLEXITY

Specifying the exact complexity of the method is difficult, As this is a non-convex optimization problem, However, we can discuss the following approximate complexity analysis. Assuming constant input and output dimensions and a given network architecture, we can take a general view on the complexity of each iteration by the batch size $m$. The heaviest computational operations at each iteration are the nearest-neighbors search, the QR decomposition and the loss computation (i.e., computation of the Rayleigh Quotient). For the nearest-neighbor search, we can use approximation techniques (e.g, LSH Gionis et al. (1999)) which work in almost linear complexity by $m$. A naive implementation of the QR decomposition would lead to an $\mathcal{O}(m^2)$ time complexity. The loss computation also takes $\mathcal{O}(m^2)$ due to the required matrix multiplication. Thereby, the complexity of each iteration is quadratic by the batch size. This is comparable to other approximation techniques such as LOBPCG Benner & Mach (2011) (which also utilizes sparse matrix operations techniques for faster implementation). However, GrEASE leverages stochastic training, allowing each iteration to consider only a batch of the data, rather than the entire dataset.

Assessing the complexity of each epoch is now straightforward, and results a time complexity of $\mathcal{O}(nm)$, where $n$, the number of samples, satisfies $n \gg m$. This indicates an almost-linear complexity.

### C.2  GRAPH CONSTRUCTION

To best capture the structure of the input manifold $\mathcal{D}$, given by a finite number of samples $\mathcal{X}$, we use a similar graph construction method used by Gomez et al. in UMAP (McInnes et al., 2018), proven to capture the local topology of the manifold at each point. However, as opposed to the method in (McInnes et al., 2018), GrEASE does not compute the graph of all points, which can lead to scalability hurdles and impose significant memory demands. Instead, GrEASE either computes small graphs on each batch, or can be provided by the user with an affinity matrix $W$ corresponding to $\mathcal{X}$. Our practical construction of the graph affinity matrix $W$ is as follows:

Given a distance measure $\delta$ between points, we first compute the $k$-nearest neighbors of each point $x_i$ under $\delta$, $\{x_{i_1}, \ldots, x_{i_k}\}$, and denote

$$\rho_i = \min_j \delta(x_i, x_{i_j}), \; \sigma_i = \text{median}\{\delta(x_i, x_{i_j}) | 1 \leq j \leq k\}$$

Second, we compute the affinity matrix using the Laplace kernel

$$W_{ij} = \begin{cases} \exp\left(\frac{\rho_i - \delta(x_i, x_j)}{\sigma_i}\right) & x_j \in \{x_{i_1}, \ldots, x_{i_k}\} \\ 0 & \text{otherwise} \end{cases}$$

Third, we symmetries $W$ simply by taking $\frac{W + W^T}{2}$.

We refer the reader to McInnes et al. (2018) for further discussion about the graph construction.

# D  GRASSMANN SCORE

In this section, we provide the formulation for the Grassmann Score (GS) evaluation method, and present simple examples to visualize its meaning.

## D.1  FORMALIZATION OF GS

First, we remind Grassmann distance (see Def. 1). Grassmann distance is a metric function between equidimensional linear subspaces, where each is represented by an orthogonal matrix containing the basis as its columns. In other words, this is a metric which is invariant under multiplication by an orthogonal matrix.

**Definition 1.** *Given two orthogonal matrices $A, B \in \mathbb{R}^{n \times k}$, the Grassmann Distance between them is defined as:*

$$d_{Gr}(A, B) = \sum_{i=1}^{k} sin^2 \theta_i$$

*where $\theta_i = \arccos \sigma_i(A^T B)$ is the ith principal angle between A and B, and $\sigma_i$ is the ith smallest singular value of $A^T B$.*

Assuming we are given a dataset $\mathcal{X} = \{x_1, \ldots, x_n\} \subseteq \mathbb{R}^d$ and a corresponding low-dimensional representation $\mathcal{Y} = \{y_1, \ldots, y_n\} \subseteq \mathbb{R}^k$. We want to evaluate the dissimilarity between the *global structures* of $\mathcal{X}$ and $\mathcal{Y}$. We build graphs from $\mathcal{X}$ and $\mathcal{Y}$, saved as affinity matrices $W_{\mathcal{X}}$ and $W_{\mathcal{Y}}$, respectively. We construct the corresponding Unnormalized Laplacians (see Sec. 3.1) $L_{\mathcal{X}}$ and $L_{\mathcal{Y}}$. We define the matrices $V_{\mathcal{X}}, V_{\mathcal{Y}} \in \mathbb{R}^{n \times t}$ so that their columns are the first $t$ eigenvectors of $L_{\mathcal{X}}, L_{\mathcal{Y}}$, respectively.

Finally, we define the GS of $\mathcal{Y}$ (w.r.t $\mathcal{X}$) as follows:

**Definition 2.** $GS_{\mathcal{X}}(\mathcal{Y}) = d_{Gr}(V_{\mathcal{X}}, V_{\mathcal{Y}})$

$t$ is a hyper-parameter of GS. A reasonable choice would be to take $t = 2$, which is equivalent to measure the Grassmann distance between the Fiedler vectors of the Laplacians. The Fiedler vector is known for its hold of the most important global properties. The larger $t$, the more complicated structures are taken into consideration in the GS computation (which is not necceray desired).

Note that for the construction of the affinity matrices $W_{\mathcal{X}}, W_{\mathcal{Y}}$ we use the same construction scheme detailed in App. C.2. This construction method is similar to the one presented by McInnes et al. (2018), and proved to capture the local topology of the underlying manifold.

It is important to note that GS might ignore the local structures, while concentrating on the global structures (especially for smaller values of $t$). The ultimate goal in visualization is to find a balance between the global and local structure.

## D.2  ADDITIONAL GS EXAMPLES

Fig. 6 depicts two additional demonstrations of the alignment between the intuitive expectation and the GS results on two toy dataset. The basic global structure of both of these datasets is two distinct clusters. This structure is indeed captured by GS. However, KNN gives perfect score also when the one of the clusters is separated. Silhouette score favourites the 2-points embedding. Namely, it trade-offs local structure (i.e., giving lower score for preserving local structure, even when the global properties are the same).

# E  ADDITIONAL REAULTS

the full results of Fig. 5a are summarized in Tab. 2.

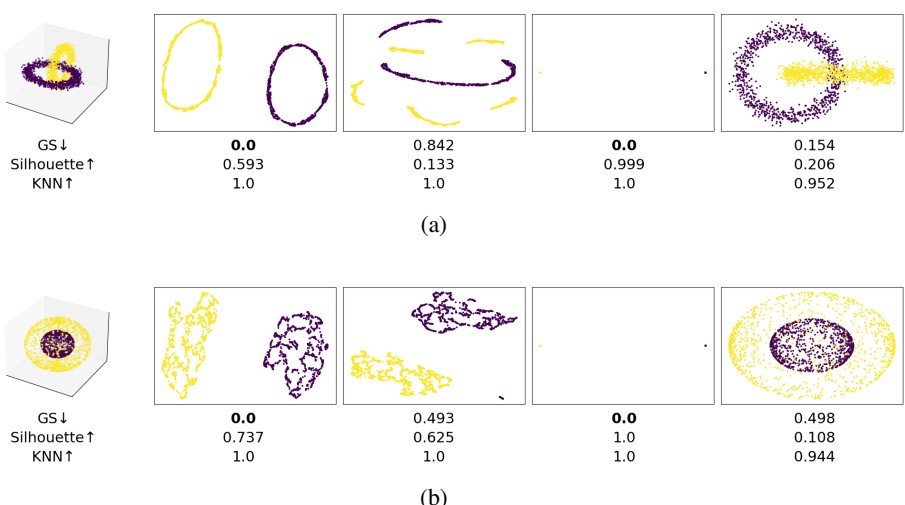

Figure 6: Additional demonstrations of the alignment between the intuitive expectation and the GS results on two toy dataset. Four possible 2-dimensional embeddings of these dataset are provided, along with their corresponding GS, KNN accuracy and Silhouette score. Unlike KNN and Silhouette, GS effectively captures the preservation of global structure.

Table 2: A comparison between GrEASE and SpectralNet dimensional SE and Fiedler Vector (FV) approximation on real-world datasets. The values are the mean and standart deviation of the $sin^2$ distance between the predicted and true eigenvector, over 10 runs. Lower is better. GrEASE ability to separate the eigenvectors is evident.

| Dataset | Method | $v_2$ | $v_3$ | $v_4$ | $v_5$ |
|---|---|---|---|---|---|
| Cifar10 | GrEASE | $0.016_{\pm 0.004}$ | $0.052_{\pm 0.008}$ | $0.069_{\pm 0.034}$ | $0.106_{\pm 0.037}$ |
| | SpectralNet | $0.449_{\pm 0.199}$ | $0.325_{\pm 0.148}$ | $0.399_{\pm 0.194}$ | $0.414_{\pm 0.17}$ |
| Appliances | GrEASE | $0.063_{\pm 0.002}$ | $0.094_{\pm 0.007}$ | $0.109_{\pm 0.001}$ | - |
| | SpectralNet | $0.307_{\pm 0.047}$ | $0.530_{\pm 0.114}$ | $0.401_{\pm 0.106}$ | - |
| KMNIST | GrEASE | $0.0.044_{\pm 0.002}$ | $0.101_{\pm 0.010}$ | - | - |
| | SpectralNet | $0.372_{\pm 0.174}$ | $0.396_{\pm 0.137}$ | - | - |
| Parkinsons | GrEASE | $0.056_{\pm 0.006}$ | - | - | - |
| | SpectralNet | $0.229_{\pm 0.138}$ | - | - | - |

## F  FINE-TUNING GrEASE WITH UMAP LOSS

One way to get a generalizable version of UMAP may be an extension of GrEASE by fine-tuning the network with UMAP loss. We tried that idea, but were forced to stop this direction, as we stumbled upon the well-known catastrophic forgetting case.

Figure 7 presents an experiment on the simple 2circles dataset. Each row is represented the same experiment, run with a different seed. We trained GrEASE to output the 2D SE of the 2circles dataset, as shown in the left column. Then, we initialized a new network, with the same architecture, with the pre-trained weights from GrEASE. This network was trained with UMAP loss, as in (Sainburg et al., 2021). We tried different learning-rates for fine-tuning, to best match the desired UMAP embedding (i.e. retaining the local structure), without losing the global structure (e.g., separation of the two clusters). Unfortunatly, there was no learning-rate that matched our goals.

## G  TECHNICAL DETAILS

To compute the ground truth SE on the train set and its corresponding eigenvalues, we constructed an affinity matrix $W$ from the train set (as detailed in Appendix C.2), with a number of neighbors detailed in Table 4. After constructing $W$, we computed the leading $k$ eigenvectors of its corresponding Unnormalized Laplacian $L = D - W$ via Python's Numpy SVD or Scipy LOBPCG SVD

Figure 7: The catastrophic forgetting phenomenon when fine-tuning GrEASE to much UMAP performance on the 2circles dataset. Each column represents a fine-tuning using a different learning-rate. Each row is a repetition, initialized with a different seed.

Table 3: Technical details of the real-world datasets used for GrEASE and NUMAP experiments.

|          | Cifar10 | Appliances | KMNIST | Parkinsons | Wine | Banknote |
|----------|---------|------------|--------|------------|------|----------|
| #samples | 60,000  | 19735      | 70,000 | 5875       | 178  | 1372     |
| #features| 500     | 28         | 784    | 19         | 13   | 4        |

(depending on the size). To get the generalization ground truth, we constructed an affinity matrix $W$ from the train and test sets combined, computed the leading $k$ eigenvectors of its corresponding Unnormalized Laplacian $L = D - W$, and extracted the representations corresponding to the test samples. We used a train-test split of 80-20 for all datasets.

For the SE implementation via sparse matrix decomposition techniques, we used Python's sklearn.manifold.SpectralEmbedding, using a default configuration (in particular, 10 jobs, 1% neighbors).

The architectures of GrEASE's and SpectralNet's networks in all of the experiments were as follows: size = 128; ReLU, size = 256; ReLU, size = 512; ReLU, size = $k+1$; orthonorm. NUMAP's second NN and PUMAP's NN architectures for all datasets was: size = 200; ReLU, size = 200; ReLU, size = 200; ReLU, size = 2; The SE dimensions for NUMAP were: Cifar10 - 20; Appliances - 10; Wine - 10; Banknote - 3.

The learning rate policy for GrEASE and SpectralNet is determined by monitoring the loss on a validation set (a random subset of the training set); once the validation loss did not improve for a specified number of epochs, we divided the learning rate by 10. Training stopped once the learning rate reached $10^{-7}$. In particular, we used the following approximation to determine the patience epochs, where $n$ is the number of samples and $m$ is the batch size: if $\frac{n}{m} \leq 25$, we chose the patience to be 10; otherwise, the patience decreases as $\max\left(1, \frac{250m}{n}\right)$ (i.e., the number of iterations is the deciding feature).

To run UMAP, we used Python's umap-learn implementation (UMAP's formal implementation). We used the built-in initialization option "spectral" (i.e., SE), and initialized contumely with PCA (implemented via Python's sklearn.decomposition.PCA) and GrEASE. For Parametric UMAP we used the Pytorch implementaion (Liu, 2024). For all methods we used a default choice of 10 neighbors.

As for the evaluation methods, we used a default choice of 5 neighbors to compute the KNN accuracy. The graph construction for GS is as detailed in App. C.2, using 50 neighbors to ensure connectivity.

Table 4: Technical details in the GrEASE experiments for all datasets.

|            | Moon      | Cifar10   | Appliances | KMNIST    | Parkinsons |
|------------|-----------|-----------|------------|-----------|------------|
| Batch size | 2048      | 2048      | 2048       | 2048      | 512        |
| n_neighbors| 20        | 20        | 20         | 20        | 5          |
| Initial LR | $10^{-2}$ | $10^{-2}$ | $10^{-3}$  | $10^{-3}$ | $10^{-2}$  |
| Optimizer  | ADAM      | ADAM      | ADAM       | ADAM      | ADAM       |

We ran the experiments using GPU: NVIDIA A100 80GB PCIe; CPU: Intel(R) Xeon(R) Gold 6338 CPU @ 2.00GHz;

