# OpenReview forum: "GrEASE: Generalizable Spectral Embedding with an Application to UMAP"
_ICLR.cc/2025/Conference — Submitted to ICLR 2025_

### Official Review · Reviewer_WUZ1 · 2024-10-16

**Soundness:** 1
**Presentation:** 2
**Contribution:** 1
**Rating:** 3
**Confidence:** 5

**Summary:**

This paper considers a parameterized technique for computing the spectral embedding (SE) of a dataset. They note that parameterized methods exist for finding an embedding with respect to spectral clustering (namely, SpectralNet). Therefore, the authors propose a modification to this technique which ensures that the eigenvectors of the Laplacian are correctly distinguishable in the obtained embedding. The authors show this to be the case experimentally and propose a metric -- the Grassmann score -- with which they measure the resulting embeddings.

**Strengths:**

I agree with and commend this paper's stated goals -- improving data visualization techniques' scalability and generalizability is a useful direction which aligns well with this conference's scope. Furthermore, I believe that parameterized methods for data visualization are under-studied and that there are interesting objectives which can be optimized in this space.

Additionally, I am interested in understanding how to produce UMAP outputs and what the properties of these outputs are. To this end, I believe the idea of the Grassmann score for measuring the quality of an embedding is an interesting one. I have long considered that the usage of kNN scores is insufficient for measuring data-visualization quality.

**Weaknesses:**

Unfortunately, the weaknesses in this paper significantly outweigh its strengths. These are organized into the following three themes: the premise of the paper, the experiments, and the technical clarity.

## Premise

Unless I am missing something, this paper's primary technical contribution is to apply a change-of-basis to the output of SpectralNet. This... does not feel like a sufficient contribution for ICLR. Furthermore, it is wholly unclear to me why this change-of-basis is even necessary. Yes -- the authors show that SpectralNet's outputs are not unique up to orthogonal transformation. And yes -- these embeddings might not correspond to the exact eigenvectors of the Laplacian. However, I fail to see how this is cause for any concern or, indeed, why this is something that needs to be remedied.

As far as I understand, the embeddings from SpectralNet and the embeddings from GrEASE are the exact same output except that GrEASE takes the points and rotates them. If we are simply rotating the outputs of SpectralNet, then how does this have any impact on downstream uses of the embeddings? For example, whether this rotation was applied clearly makes no difference if we then use the embedding as an initialization for UMAP (as is the primary use-case in the paper) or for spectral clustering (the most common use for spectral embeddings). Thus, I find it difficult to understand the primary motivation for this technical contribution in the first place. Please correct me if I have misunderstood something.

## Experiments

Part of the confusion surrounding the motivation comes down to the experiments: it seems that the authors have chosen experiments which emphasize that GrEASE finds the exact eigenvectors of the Laplacian but have not run experiments to explain why this was necessary to do. For example, the usage of the Grassmann score to measure distance between the Laplacian vectors and the identified vectors shows that GrEASE is better at identifying the eigenvectors of the Laplacian. However, this experiment does little to explain *why* this is helpful. Instead, the only evidence of GrEASE's improved embeddings seems to be Figure 1, where results are shown on three toy datasets. The sparsity of the dataset choices and the lack of quantitative measures however does not make Figure 1 sufficient evidence to justify the fact that GrEASE was necessary (additionally, there is no code attached so the experiments are not reproducible).

Instead, I would be more interested in an in-depth evaluation of the outputs of GrEASE. The authors have proposed a dimensionality reduction technique for getting a spectral embedding and yet we do not see a single spectral embedding in the paper. What do the GrEASE outputs look like? How do they compare against standard Laplacian Eigenmaps or spectral embeddings? Similarly, the paper is in need of a comparison between SpectralNet and GrEASE outputs. If the authors are crafting their submission around modifications to SpectralNet, then a comprehensive evaluation of why this was necessary is in order. Are GrEASE outputs significantly different from SpectralNet ones? An experimental verification of the issues of SpectralNet outputs would be helpful in explaining why GrEASE was necessary.

## Technical Clarity

These issues are exacerbated by the writing of the paper. While the text is often easy to read from a stylistic perspective, the technical substance is imprecise and the reader must make guesses about what the authors mean.

The most pertinent example of this is the term "eigenvector separation," which is used throughout the paper. This is not a term that I am familiar with and it took until page 5 that I even had a guess as to what was meant by it (I believe it means that the spectral embedding is aligned with the eigenvectors rather than some linear combination of them). Since this is a key notion behind the paper, it requires a clear, unambiguous definition.

In addition to this, it seems that the term "spectral embedding" is used in a specific way which feels different from my expectation. I would expect the spectral embedding to be the $n \times k$ matrix of leading eigenvectors of the Laplacian *or any orthogonal transformation of this matrix*. Indeed, the orthogonal transformation is arbitrary with respect to the relevant use-cases of spectral embeddings such as spectral clustering and UMAP. Thus, the implication in lines 239-240 that "this does not lead to the [spectral embedding]" feels unsubstantiated.

The writing has several smaller cases of impreciseness. For example, the first sentences of the intro state that SEs are used in GNNs and GCNs? I don't believe this to be true. Additionally, the Grassmann Score is left completely undefined and I have no idea how one calculates it. Furthermore, the applications seem incomplete. It is currently completely unclear to me how GrEASE relates to Diffusion Maps. Separately, if one wished to approximate the Fiedler vector, then this is a well-studied problem and there are fast methods for doing it such as [1]. Lastly, I strongly disagree that the complexity is linear in the number of samples (line 315). For one thing, I'm not sure what is meant by "samples" here. More importantly, however, I think Figure 2 is comparing apples to oranges. GrEASE works by running mini-batches and averaging their outputs. However, it seems the competitors are running on all $n$ inputs? Also, are these all on GPU or are we comparing CPU times to GPU times? Does the runtime include training time? How many batches were run? If the code was available, these things would be easier to check.


[1]: Fast and Simple Spectral Clustering in Theory and Practice; Peter Macgregor at NeurIPS 2023

**Questions:**

My primary question is for the authors to explain why GrEASE is necessary as a substitute for SpectralNet. Specifically,
- What happens if we use SpectralNet outputs as the input to Parametric UMAP (rather than GrEASE outputs)? I suspect that this will perform equivalently to NUMAP and would require thorough, reproducible experiments to convince me otherwise. If SpectralNet and GrEASE perform the same as an input to UMAP, this would contradict the primary motivation of the paper that the eigenvector separation is indeed necessary.
- How do GrEASE and SpectralNet outputs differ? How do they compare to standard Laplacian Eigenmaps/spectral embeddings?
- How does GrEASE help in getting generalizable diffusion maps? This is completely unclear to me.

---

> ### Author Response · Authors · 2024-11-22
> **Response to reviewer (part 1/2)**
>
> Thank you for finding our work's goal important and relevant, and encouraging the use of our proposed evaluation metric - Grassmann Score. We address you concerns below.
>
> **Questions**
>
> **Comment 1:** What happens if we use SpectralNet outputs as the input to Parametric UMAP (rather than GrEASE outputs)? I suspect that this will perform equivalently to NUMAP and would require thorough, reproducible experiments to convince me otherwise. If SpectralNet and GrEASE perform the same as an input to UMAP, this would contradict the primary motivation of the paper that the eigenvector separation is indeed necessary.
>
> **Response:** NUMAP heavily relies on GrEASE's ability to output the separated eigenvectors. A key architectural feature of NUMAP is the residual connections, which bias the NUMAP embeddings to stay close to the 2-dimensional SE. The second step of NUMAP involves learning the embedding from the $\ell$-dimensional SE ($\ell > 2$). GrEASE plays a crucial role in enabling the effective use of these residual connections, as without it, NUMAP embeddings would not be biased toward preserving the graph's global structure.
>
> **Comment 2:** How do GrEASE and SpectralNet outputs differ? How do they compare to standard Laplacian Eigenmaps/spectral embeddings?
>
> **Response:** This is extensively shown in Sec. 5.
>
> **Comment 3:** How does GrEASE help in getting generalizable diffusion maps? This is completely unclear to me.
>
> **Response:** GrEASE enables the output of both eigenvalues and eigenvectors, which is not possible with SpectralNet. This allows for the multiplication of each eigenvector by its corresponding eigenvalue, resulting in the Diffusion Maps embedding.
>
> **Weaknesses**
>
> **Comment:** Unless I am missing something, this paper's primary technical contribution is to apply a change-of-basis to the output of SpectralNet. I fail to see how the eigenvector separation is cause for any concern, and why this is something that need to be remedied.
>
> **Response:** Eigenvector separation (achieved by the change-of-basis operation) is crucial for various applications of SE, such as the Fiedler vector, eigenvalues approximation, and Diffusion Maps, which are discussed in Section 4.4. Specifically, we focus on SE application to visualization, via NUMAP.
> NUMAP relies substantially on the eigenvectors separation for its residual connections, which are designed to bias NUMAP embeddings towards the 2-dimensional SE (learned from $\ell$-dimensional SE). Previous works have explored (infeasible) solutions to the eigenvector separation challenge (e.g., [1]-[4]).
>
> Additionally, change of basis is the key of many important algorithms in machine learning (e.g., PCA, CCA, Fourier transform, Wavelet transform). Solving the eigenvector separation problem is, by definition, a change of basis operation. Hence, we disagree with the reviewer claim that a change-of-basis is not enough of a contribution.
>
> **Comment:** The authors used Grassmann Score to measure distance between the Laplacian vectors and the identified vectors.
>
> **Response:** We would like to emphasize that the Grassmann Score is not intended to evaluate the quality of eigenvector separation and was not used or presented as such in the paper. Instead, the Grassmann Score is suitable for assessing graph-based dimensionality reduction methods, where the objective is to preserve the graphical structure of the representations.

---

> > ### Comment · Reviewer_WUZ1 · 2024-11-24
> >
> > I understand that the authors strongly disagree with my review, but I promise my comments are purely meant to be constructive. I start with one big-picture point and then respond to a few of the smaller points after that in the next comment.
> >
> > -----
> >
> > ### Big Picture
> >
> > To be clear, I am 100% on board with the idea of Lemma 1 -- the solutions to the Rayleigh quotient are equivalent up to an orthogonal transformation. This is based on the well-known fact that the trace is invariant to the chosen basis [1]. As a result, this holds true for all dimensionality reduction techniques which rely on the trace of a matrix such as LLE [2, Eq. 14], Laplacian Eigenmaps [3, shortly after Eq. 3.1], and PCA [2] (to name a few). In some sense, this is inevitable -- if we are interested in preserving the pairwise relationships between points, then we do not care how these points are rotated.
> >
> > So I fully agree with the authors on this fact. What I am struggling to see (and what the authors' comments/paper have not convinced me of) is why this is a problem that needs solving. If I am using the embedding for clustering (as in spectral clustering) or initialization (as in UMAP/P. UMAP), it is irrelevant whether the embedding is rotated.
> >
> > Furthermore, regarding this being a sufficient contribution, I'd point out that change-of-basis is a standard notion and I am not questioning its utility. I'm instead questioning whether modifying an algorithm by changing its output basis is sufficiently novel for ICLR.
> >
> > Regarding the point that NUMAP uses residual connections, I'm sorry but there is not sufficient context to know what the authors mean by this. First, the code is still not available as supplementary material so I am not able to run any of this and see what is meant. Second, the description in the paper simply states:
> >
> >     To enhance this capability, we incorporate residual connections from the first to the last layer of the second NN. Specifically, the objective is to minimize the residual between the ℓ-dimensional UMAP embedding and the ℓ-dimensional SE.
> >
> > But... what is the objective function? What is the training process? I know there are more details in the appendix but those do not explain to me how this set up either. I'll restate my confusion again -- if the embedding from GrEASE is a rotated version of the embedding from SpectralNet, how could it change to what extent the global structure is preserved? Why can one not simply add the residual connections to P. UMAP? Wouldn't this train P. UMAP to preserve the rotated version of the input? Being precise about the writing and including experiments which evaluate all of the authors' proposed changes would go a long way to clearing up this kind of confusion.
> >
> > As said in the original review, in order to convince me that it'll make a difference, I'd need to see experiments showing (a) what the output of GrEASE vs. SpectralNet (SN) is across many datasets and hyperparameters, (b) what the output of P. UMAP is if connected to GrEASE or SN, and (c) what the output of NUMAP is when connected to GrEASE or SN. I would also need to see the code in order to reproduce these experiments (I've actually tried reproducing the P. UMAP column of Figure 1 and have not been able to).
> >
> > [1]: https://math.stackexchange.com/a/165845/432295
> > [2]: https://arxiv.org/pdf/2011.10925
> > [3]: https://www2.imm.dtu.dk/projects/manifold/Papers/Laplacian.pdf
> > [4]: https://stats.stackexchange.com/a/136072/173038

---

> ### Author Response · Authors · 2024-11-22
> **Response to reviewer (part 2/2)**
>
> **Comment:** The term "eigenvector separation" is unknown and undefined.
>
> **Response:** The term "eigenvector separation" is a key concept in the paper and was clearly defined in lines 44-45 at the beginning of the document. Additionally, this term is used in the literature (see e.g., [1]).
>
> **Comment:** I don't believe the authors claim that SE is used in GNNs and GCNs.
>
> **Response:** In the paper, we support the claim that SE is used in GNNs and GCNs with citations to three papers [5]-[7]. For brevity, we summarize the use of SE in these methods: Eigen-GNN [5] integrates SE with GNNs to reduce the feature-centricity of modern GNNs and enable learning on the graph structure domain; [6] uses SE to construct globally consistent anisotropic kernels for GNNs, enabling graph convolutions based on topologically-derived directional flows; [7] utilizes SE to design fast localized GCNs.
>
> **Comment:** The Grassmann Score is left completely undefined and I have no idea how one calculates it.
>
> **Response:** As referred in lines 419-420, the Grassmann Score is formally define in App. D, which also includes the method for calculating GS.
>
> **Comment:** I strongly disagree with the statement line 315, that the complexity is linear in the number of samples. Also, I think Fig. 2 is comparing apples and oranges.
>
> **Response:** We disagree with the author's statement. Line 315 is supported by Figure 2b and a detailed discussion in Appendix C. In Appendix C, we provide a quick complexity analysis, including training time, and demonstrate that the complexity is linear with respect to the number of samples. Figure 2b includes a simple empirical experiment comparing GrEASE's complexity to non-parametric SE implementations, which also supports linear complexity. Of course, GrEASE's times account for both training and inference on all inputs. The non-parametric methods, lacking GPU support, were run on CPU with multiple jobs. The technical details (e.g., number of jobs, batch sizes, etc.) are provided in Appendix G.
>
> **Comment:** if one wished to approximate the Fiedler vector, then this is a well-studied problem and there are fast methods for doing it such as [8].
>
> **Response:** It seems that the cited paper only enables fast approximation of the Fiedler vector, without supporting generalization to test samples. This is a special case of the "Generalizable and Eigenvector Separation" paragraph in the Related Work section (Sec. 2). In contrast, GrEASE enables generalizable Fiedler vector approximation, as discussed in Section 4.4 and empirically demonstrated in Section 5.1.
>
> [1] D. Pfau et. al. , “Spectral inference networks: Unifying deep and spectral learning,” ICLR, 2019.
>
> [2] I. Gemp et. al, “Eigengame: Pca as a nash equilibrium,” ICLR, 2021.
>
> [3] Z. Deng et. al, “Neuralef: Deconstructing kernels by deep neural networks,” ICML, 2022.
>
> [4] Z. Chen et. al, “Specnet2: Orthogonalization-free spectral embedding by neural networks,” MSML, 2022.
>
> [5] Z. Zhang et. al, “Eigen-gnn: A graph structure preserving plug-in for gnns,” IEEE, 2021.
>
> [6] D. Beaini et. al, “Directional graph networks,” ICML, 2021.
>
> [7] M. Defferrard et. al, “Convolutional neural networks on graphs with fast localized spectral filtering,” NeurIPS, 2016.
>
> [8] P. Macgregor, “Fast and simple spectral clustering in theory and practice,” NeurIPS, 2023.

---

> > ### Comment · Reviewer_WUZ1 · 2024-11-24
> >
> > Continuation of my response to the comments (in no specific order)
> >
> > -----
> >
> > ### Other Points
> >
> > - Regarding how GrEASE and SN output differ, I'm sorry -- I don't see it in Section 5. What I see in Section 5 is the fact that GrEASE identifies the true eigenvectors whereas SN finds the basis of the eigenvectors. Since GrEASE involves a change-of-basis operation onto the Laplacian's spectrum, these experiments fall into the category of "verifying that it works". But, because I am unconvinced that preserving these eigenvectors is beneficial, I am missing a comparison of GrEASE and SpectralNet outputs that would convince me that this was worth doing. For example, I'd be interested in things such as (a) how these embeddings look, (b) quantitative measures for evaluating this (beyond the GS which records that the eigenvectors are separated as one should assume they will be after a change-of-basis), and (c) all of this across a wide variety of datasets. These experiments should serve to convince the reader that the GrEASE outputs provide something which the SpectralNet outputs do not.
> >    - Put another way, the paper says "one needs to find the basis of the spectral embedding" and then says "using a change-of-basis, we find that basis". But since I am not convinced that one needs the basis for downstream use-cases, the paper's current experiments are largely missing the mark for me.
> > - Regarding the GS, I'm sorry again for being difficult, but... isn't it directly being used to measure eigenvector separation? Columns 2-4 of Figure 2 and Figure 5a all show the $\sin ^2$ value between vectors. Isn't this exactly a subset of the sum for the GS?
> >    - Also, I should have been clearer in the original point in my review -- the GS is undefined in the main text and it's therefore not clear how to calculate it from reading the main paper.  As stated in the ICLR call for papers, papers should be self-contained to the main text. I just meant this as an example of the main text's language being imprecise.
> > - Regarding the definition of eigenvector separation -- I apologize. I somehow missed it in the introduction during my multiple read-throughs.
> >    - But also, the term "eigenvector separation" is not used anywhere in [1]? They do discuss separating eigenvectors (in Appendix C.1) but it seems they're using it in a different context -- their goal seems to be to ensure orthogonality between the vectors they find. The authors of the present paper, instead, are implying a change-of-basis to the eigenvectors.
> > - For the point on GNNs and GCNs, I agree that there are cases where one may use spectral embeddings in GNNs and GCNs and that people have likely tried this. But this wasn't the point I was trying to make (I again should have been more precise with my comment). I meant that simply stating in the introduction that "Notable applications [of spectral embeddings] include GNNs, GCNs" is misleading because spectral embeddings are not used in these methods in their general, well-known variants. Thus, I meant that I am not convinced by this statement as it is phrased and that this was therefore an example of imprecise writing.
> >    - I know we're really getting into nitpicking here (and I apologize) but I ended up looking through the references to double-check whether spectral embeddings are used in GNNs and GCNs and I had just misunderstood these techniques. It seems that some of the references don't discuss spectral embeddings at all? Namely, the papers [2-4] don't seem to use spectral embeddings in them but are cited as examples of SE in GCNs, etc. This is kind of the point I was trying to make -- the default versions of these tools do not rely on spectral embeddings in the way that the sentences in the introduction seem to imply.
> > - Lastly, let's discuss the runtime. The standard notion of time-complexity is a worst-case setting. The authors in Appendix C state that the time-complexity is linear per-epoch (I'll ignore the fact that this analysis overlooks the number of parameters in the neural network, which should get factored into the big-O notation). So, how many epochs are needed to train this? I'd argue that this is dataset-dependent. Since time-complexity is explicitly a worst-case evaluation, the runtime is therefore *not* linear in the number of samples. It is $O(n\mathcal{E})$, where $\mathcal{E}$ is the number of training epochs. The fact that the authors chose a simple dataset (2-moons) where few epochs are needed to demonstrate time-complexity being linear is not evidence that it is linear across all settings.
> >    - Also, one really can't use it as evidence that a method is faster if it ran on GPU and the competitors were on CPU... (especially without mentioning this in the main text)
> >
> > [1] https://openreview.net/pdf?id=SJzqpj09YQ
> > [2] https://arxiv.org/pdf/1606.09375
> > [3] https://arxiv.org/pdf/2012.09699
> > [4] https://arxiv.org/pdf/2106.03893

---

### Official Review · Reviewer_y9N7 · 2024-10-28

**Soundness:** 3
**Presentation:** 2
**Contribution:** 2
**Rating:** 3
**Confidence:** 4

**Summary:**

The authors propose a new method that aims to achieve a three-fold contribution to the field of dimensionality reduction: (1) parametric embedding, (2) eigenvector separability, and (3) output of eigenbasis.

They combine this approach with UMAP, leading to another method called NUMAP.

**Strengths:**

The authors provide a good framework in which to embed their own work.

The related work section seems extensive.

The appendix appears extensive.

From their claims it seems promising that a new approach can strike a balance between the three described SE goals.

**Weaknesses:**

If, as described in ll. 341ff, the representation is learned from the UMAP embedding space via a parametric network, then it is unclear how this approach can have more scalability than parametric UMAP (which only does the first part), which the authors claim in their introduction.

I am surprised to see that the time for computing the spectral embedding via the eigensolvers AMG and LOBPCG are so slow.  Could this perhaps be related to the construction of the kNN graph?  If so, it would be better to either exclude the graph contstruction or to use the same graph building as UMAP does (which uses an approximate version via annoy or pynndescent).

Overall, it feels like it's more of a minor extension of Shaham et al. (2018) and it is not sufficiently clear why the improvements are significant enough.  This also relates to a comparison that is mostly only done to parametric UMAP, raising concerns about the tanglible improvements over prior work.

The choice of using the Grassmann score seems unusual.  Why are other approaches like correlation between high-dimensional and low-dimensional points not considered?  Especially since it's the main metric that they show improvements on, it feels like it is not discussed enough in the main text.

**Questions:**

- Why is outputting the basis relevant?  The authors give a few citations, but it would be good to highlight that in their own words.
- Why are the visualizations in 5b only compared to parametric UMAP?  It would be good to see how other approaches fare on the data.
- Please make sure that the scatter plot visualizations have a 1:1 aspect ratio.  They look very distorted in e.g. Fig. 7, I am unsure whether this is then also distorted in the main body figures.
- The embeddings in Fig. 4 are not labeled.
- Since the authors talk a lot about scalability, it would be nice to see some large-scale real-world datasets
- What is meant by UMAPs contrastive loss (l. 343)?  UMAP uses a binary cross-entropy loss.
- It would be good to make the evaluation more consistent.  Sometimes only GS and kNN are reported, and sometimes the silhouette score.  It would be beneficial to include all of the metrics and summarize them in one place.

---

> ### Author Response · Authors · 2024-11-22
> **Response to reviewer**
>
> Thank you for finding our work extensive and promising for future work to rely on. We address you concerns below.
>
> **Questions**
>
> **Comment 1:** Why is outputting the basis relevant? The authors give a few citations, but it would be good to highlight that in their own words.
>
> **Response:** Eigenvector separation is crucial for various applications of SE, such as the Fiedler vector, eigenvalues approximation, and Diffusion Maps, which are discussed in Section 4.4. Specifically, we focus on SE application to visualization, via NUMAP.
> NUMAP relies substantially on the eigenvectors separation for its residual connections, which are designed to bias NUMAP embeddings towards the 2-dimensional SE (learned from $\ell$-dimensional SE). Previous works have explored (infeasible) solutions to the eigenvector separation challenge (e.g., [1]-[4]).
>
> **Comment 2:** Why are the visualizations in 5b only compared to parametric UMAP? It would be good to see how other approaches fare on the data.
>
> **Response:** As of today, UMAP is the most popular visualization technique, and thus it serves as our baseline. The only parametric version of UMAP we are familiar with is Parametric UMAP, which we use for comparison with NUMAP.
>
> **Comment 3:** Please make sure that the scatter plot visualizations have a 1:1 aspect ratio. They look very distorted in e.g. Fig. 7, I am unsure whether this is then also distorted in the main body figures.
>
> **Response:** This is only relevant to Fig. 7. Thank you for pointing this out.
>
> **Comment 4:** The embeddings in Fig. 4 are not labeled.
>
> **Response:** Fig. 4 is dedicated to understanding of Grassmann Score (GS). Specifically, it compares four embeddings using KNN, Silhouette and GS. The embedding methods used to create these embedding are not relevant to the analysis.
>
> **Comment 5:** Since the authors talk a lot about scalability, it would be nice to see some large-scale real-world datasets
>
> **Response:** We present results on relatively large-scale datasets (CIFAR10 and KMNIST) in Sec. 5.1, where computing the ground-truth SE is feasible. Additionally, we conduct a scalability experiment in Sec. 5.2 to assess the runtime performance on large-scale datasets.
>
> **Comment 6:** What is meant by UMAPs contrastive loss (l. 343)? UMAP uses a binary cross-entropy loss.
>
> **Response:** We view binary cross-entropy loss as a contrastive loss, in the sense that it aims to bring paired points closer while pushing other random points apart. That is, for instance, different from the SE objective, which seeks to bring similar points closer while enforcing identity covariance.
>
> **Comment 7:** It would be good to make the evaluation more consistent. Sometimes only GS and kNN are reported, and sometimes the silhouette score. It would be beneficial to include all of the metrics and summarize them in one place.
>
> **Response:** The evaluation metrics used are KNN and GS, which are summarized in Tab. 1. The Silhouette scores are included in Fig. 4 solely to highlight its insufficiency in capturing global structure preservation in dimensionality reduction.
>
> **Weaknesses**
>
> **Comment:** It is unclear how NUMAP can have more scalability than parametric UMAP (which only does the first part), which the authors claim in their introduction.
>
> **Response:** First, we did not claim that NUMAP can have more scalability than P. UMAP. However, it is important to note that their scalability is comparable, in the sense that there is no dataset size that Parametric UMAP can handle which NUMAP cannot.
>
> **Comment:** The choice of using the Grassmann score seems unusual. Why are other approaches like correlation between high-dimensional and low-dimensional points not considered?
>
> **Response:** We are not familiar with this evaluation metric and are unsure of its meaning or how it is computed. As discussed in detail in Section 4.5, we used the Grassmann Score because popular evaluation metrics (e.g., kNN and Silhouette) fail to capture global structure preservation, whereas the Grassmann Score does.
>
> [1] D. Pfau et. al. , “Spectral inference networks: Unifying deep and spectral learning,” ICLR, 2019.
>
> [2] I. Gemp et. al, “Eigengame: Pca as a nash equilibrium,” ICLR, 2021.
>
> [3] Z. Deng et. al, “Neuralef: Deconstructing kernels by deep neural networks,” ICML, 2022.
>
> [4] Z. Chen et. al, “Specnet2: Orthogonalization-free spectral embedding by neural networks,” MSML, 2022.

---

### Official Review · Reviewer_ywmX · 2024-11-01

**Soundness:** 2
**Presentation:** 3
**Contribution:** 3
**Rating:** 6
**Confidence:** 4

**Summary:**

This paper addresses limitations in Spectral Embedding (SE), a popular non-linear dimensionality reduction technique widely used in applications like visualization (e.g., UMAP), Graph Neural Networks (GNNs), and protein analysis. SE’s strengths lie in its ability to preserve global data structure through projections onto eigenvectors of the Laplacian matrix. However, SE faces three main challenges in modern applications: generalizability (handling new data points without re-computation), scalability (processing large datasets efficiently), and eigenvector separation (directly extracting leading eigenvectors).

The authors build on SpectralNet to introduce GrEASE (Generalizable and Efficient Approximate Spectral Embedding), which achieves scalability, generalizability, and eigenvector separation, enhancing SE’s utility across diverse applications. They further present NUMAP, a novel adaptation combining SE initialization with UMAP’s loss function, achieving results similar to standard UMAP while also generalizing to new data. NUMAP addresses limitations in Parametric UMAP (P. UMAP), which lacks global structure preservation.

The contributions may include:

1. GrEASE, a generalizable SE method with eigenvector separation.

2. A foundation for new SE applications and enhancements.

3. NUMAP, an adaptation of GrEASE for generalizable UMAP.

4. A new evaluation method for dimensionality reduction, enabling better assessment of global structure preservation.

**Strengths:**

1. The introduction of NUMAP as a generalizable alternative to UMAP, which preserves the global structure while allowing for out-of-sample extension, showcases creative integration of SE principles with modern visualization techniques, addressing limitations in Parametric UMAP (P. UMAP).

2. The authors address common challenges in SE with rigorous methodological enhancements, particularly with the post-processing approach for eigenvector separation, which ensures that GrEASE provides accurate and interpretable embeddings.

3. The proposed evaluation metric for global structure preservation could also become a valuable tool for assessing future dimensionality reduction techniques.

**Weaknesses:**

1.  While the paper introduces GrEASE as an enhancement to SE, the experimental comparisons focus mainly on classical SE methods and UMAP adaptations, with limited exploration of other state-of-the-art generalizable embedding methods or scalable techniques that target similar applications, such as LargeVis or PaCMAP.

2. The experiments mainly rely on synthetic and operational datasets, which may not fully capture the complexity or variability encountered in real-world applications where SE is commonly applied, such as social network analysis or biomedical data.

**Questions:**

In additional to the above comments, this reviewer does have the following specific questions on technical details.

1. Minimising R_L(U) is actually a Grassmann manifold optimization problem. The paper could elaborates on this further.  Lemma 1 is in fact an explicit fact on the Grassman manifold.

2. Between Line 249 and 251, not sure why suddenly talk about Laplace-Beltrami operator.  In the paper context, Laplacian matrix is sufficient enough.

3. In line 262, can you please more specifc on the meaning of "it becomes apparent that the SE was seldom attained."?

4. I would like to see an exact definition of the separation of the eigenvectors.

5. In line 294, why you say "Q is a property of ..."?

6.  Line 298:  Can you give a proof that the eigenvalues of \tilde{\Lambda} are the eigenvalues of L", particularly when  \tilde{\Lambda} comes from the average of batch operations?

7. Can algorithm 2 be moved into the main text?

8. In Algorithm 1 (line 754), which Sec. 3.2.  I did not find out loss function.

9. Algorithm 2 is not clearly defined.  After getting Q, how is this converted to the algorithm output F_{\theta}? Do you mean F_{\theta} = VQ, while V comes from the network output?

Overall, the paper should be carefully revised to make it more specific and concise.  For example, I would like to see paragraph between 294 and 298) to be replaced with mathematical description.

---

> ### Author Response · Authors · 2024-11-22
> **Response to reviewer**
>
> Thank you for finding our work creative and rigorous, and encouraging the use of our proposed evaluation metric - Grassmann Score. We address you concerns below.
>
> **Comment 1:** Minimising $R_L(U)$ is actually a Grassmann manifold optimization problem. The paper could elaborates on this further. Lemma 1 is in fact an explicit fact on the Grassman manifold.
>
> **Response:** Thank you for the reference. Lemma 1 is a key part of the motivation behind our approach, and for the sake of conciseness, we provide our own proof.
>
> **Comment 2:** Between Line 249 and 251, not sure why suddenly talk about Laplace-Beltrami operator. In the paper context, Laplacian matrix is sufficient enough.
>
> **Response:** The Laplace-Beltrami operator gives rise to the generalization of spectral embedding (SE). Specifically, the out-of-sample extension we seek corresponds to the leading eigenfunctions of the Laplace-Beltrami operator. While the Laplacian matrix operates solely on the training samples, the Laplace-Beltrami operator acts on the manifold (the distribution), allowing us to extend beyond the given matrix.
>
> **Comment 3:** In line 262, can you please more specifc on the meaning of "it becomes apparent that the SE was seldom attained."?
>
> **Response:** As defined in lines 172-173, the SE is the basis of the leading eigenvectors of the corresponding Laplacian matrix. Let us focus on the bottom-right histogram in Fig. 2a, which shows the distances between the true 3rd eigenvector and the predictions from SpectralNet and GrEASE. While GrEASE consistently achieves near-zero distances to the ground truth (left bin), SpectralNet does not - its distances are spread across the entire range from 0 to 1.
>
> **Comment 4:** I would like to see an exact definition of the separation of the eigenvectors.
>
> **Response:** See lines 44-45.
>
> **Comment 5:** In line 294, why you say "Q is a property of ..."?
>
> **Response:** $Q$ is manifested through the parameters of the trained network. Namely, assuming convergence to global minima, $F_{\theta}(x_i) = v_iQ$ for all $1\leq i \leq n$, where $v_i$ is the $i$th row of $V$.
>
> **Comment 6:** Line 298: Can you give a proof that the eigenvalues of $\tilde{\Lambda}$ are the eigenvalues of L", particularly when $\tilde{\Lambda}$ comes from the average of batch operations?
>
> **Response:** The proof is immediate from the equation in the bottom of page 5, by definition of $\tilde{\Lambda}$. We cannot guarantee the convergence of the average of batches. However, assuming convergence to global minima, $F_{\theta}(x_i) = v_iQ$ for all $1\leq i \leq n$, where $v_i$ is the $i$th row of $V$. That is, computing $\tilde{\Lambda}$ on a single sample would be sufficient. Since global convergence does not occur in practice, we average over multiple batches to mitigate the noise. Our experiments demonstrate the effectiveness of this method.
>
> **Comment 7:** Can algorithm 2 be moved into the main text?
>
> **Response:** Due to space limitations, we place Alg. 2 in the appendix.
>
> **Comment 8:** In Algorithm 1 (line 754), which Sec. 3.2. I did not find out loss function.
>
> **Response:** The loss function is the Rayleigh Quotient, as defined in the Sec 3.2. For brevity, we will add there the loss explicitly: $L_{\text{spectralnet}}(X, \theta) = \frac{1}{m^2}R_L\big(F_{\theta}(X)\big)$.
>
> **Comment 9:** Algorithm 2 is not clearly defined. After getting Q, how is this converted to the algorithm output $F_{\theta}$? Do you mean $F_{\theta} = VQ$, while V comes from the network output?
>
> **Response:** This is explained in lines 309-311. We will add it to algorithm 2. The idea is that we obtain $F_{\theta}$, which, assuming convergence to global minima, satisfies $F_{\theta}(X) = VQ$. After computing $Q^T$, we can multiply these representations by $Q^T$ to get the SE $F_{\theta}(X)Q^T = VQQ^T = V$

---

> > ### Comment · Reviewer_ywmX · 2024-11-26
> > **Thanks**
> >
> > Thank you for addressing my concerns and comments.  I am satisfied with your answer.  I will upgrade my rating. All the best.

---

### Official Review · Reviewer_eqZB · 2024-11-04

**Soundness:** 3
**Presentation:** 3
**Contribution:** 3
**Rating:** 5
**Confidence:** 4

**Summary:**

In this paper, the authors propose a post-processing step named GrEASE (Generalizable and Efficient Approximate Spectral Embedding) to handle three drawbacks in spectral embedding including generalizability, scalability, and eigenvectors separation.

**Strengths:**

To handle the three limitations of spectral embedding, the authors design a Generalizable and Efficient Approximate Spectral Embedding (GrEASE) method. Meanwhile, it also is extended to UMAP visualization. Besides, the presentation of the paper is good and clear to follow.

**Weaknesses:**

1) The main contribution seems to add the post-processing step of the SpectralNet. It still needs to rely on SpectralNet to solve the generalizability and scalability problems.
2) There are some problems with the proposed method including the correctness, persuasiveness, and presentation.
3) In experiments, the experiments are unclear and the evaluation is not persuasive.
4) There are some typo errors.

**Questions:**

In this paper, the authors propose a post-processing step named GrEASE and extend it to the UMAP visualization. Here are some comments:
1) As shown in introduction, the main contribution seems to add the post-processing step of the SpectralNet. It still needs to rely on SpectralNet to solve the generalizability and scalability problems. Although it can be extended to UMAP, the contribution is not enough.
2) The spectral embedding is a representation learning method. Thus, the authors could discuss the extension of the proposed method on main representation learning such as GNN or transformer.
3) In section 4.1, what are matrix Q and V? What is the dimension of V? Is it a QR decomposition of U? The authors should discuss this equation and motivation in more detail.
4) In section 4.2, why does the proposed method solve the k+1 eigenfunctions at first, instead of directly solving k eigenfunctions?
5) In line 453, what is the true vector? And how to obtain them for evaluation?
6) The representation learning method generally is evaluated on downstream tasks. Thus, the authors should add experiments about this like SpectralNet evaluated on clustering.
7) In Table 4, the batch size is so large. Thus, I'm concerned about the stability of the model. The authors should analyze this convergence.
8) There are some typo errors including missing definitions of the symbols and missing sequence numbers of the equations.

---

> ### Author Response · Authors · 2024-11-22
> **Response to reviewer**
>
> Thank you for your review. We address you concerns below.
>
> **Comment 1:** As shown in introduction, the main contribution seems to add the post-processing step of the SpectralNet. It still needs to rely on SpectralNet to solve the generalizability and scalability problems. Although it can be extended to UMAP, the contribution is not enough.
>
> **Response:** As outlined in the introduction, our two primary contributions are (1) a significant post-processing procedure and (2) NUMAP, a novel extension of GrEASE to UMAP visualization. We would like to re-emphasize the importance of eigenvector separation in GrEASE, as it underpins various SE applications. For instance, in Section 4.4, we discuss applications like the Fiedler vector, eigenvalues approximation, and Diffusion Maps. In our work, we focus particularly on the application of SE for visualization, which is achieved through NUMAP.
> NUMAP relies substantially on the eigenvectors separation for its residual connections, which are designed to bias NUMAP embeddings towards the 2-dimensional SE (learned from $\ell$-dimensional SE). Previous works have explored (infeasible) solutions to the eigenvector separation challenge (e.g., [1]-[4]).
>
> **Comment 2:** The spectral embedding is a representation learning method. Thus, the authors could discuss the extension of the proposed method on main representation learning such as GNN or transformer.
>
> **Response:** GNNs and transformers represent specific examples within the broad range of SE applications, and we see these as promising directions for future research. In this work, however,
> we focused on the UMAP extension of SE, and thoroughly presented NUMAP.
>
> **Comment 3:** In section 4.1, what are matrix Q and V? What is the dimension of V? Is it a QR decomposition of U? The authors should discuss this equation and motivation in more detail.
>
> **Response:** The matrices are clearly defined in the section (see lines 220 and 224), and we restate them here for clarity: given $n$ samples, $V\in\mathbb{R}^{n\times k}$ is the matrix constructed of the first $k$ eigenvectors of the Laplacian, $Q\in\mathbb{R}^{k\times k}$ is an arbitrary orthogonal matrix, and $U := VQ$ is an arbitrary rotation of $V$ (i.e., different from $V$, but its columns span the same space).
>
> **Comment 4:** In section 4.2, why does the proposed method solve the k+1 eigenfunctions at first, instead of directly solving k eigenfunctions?
>
> **Response:** As explained in the Sec. 3 (lines 172-173), SE is defined by the leading eigenvectors (i.e., excluding the first trivial eigenvector). Thus, GrEASE first computes the space spanned by the first $k+1$ eigenvectors, and the excludes the first, thereby outputting the leading $k$. SpectralNet, in contrast, lacks this capability.
>
> **Comment 5:** In line 453, what is the true vector? And how to obtain them for evaluation?
>
> **Response:** The "true vector" refers to the eigenvector computed using a well-studied non-parametric method (specifically, we used LOBPCG [5]), and used as the ground truth. These are used as baselines, although they lack the generalization capabilities.
>
> **Comment 6:** The representation learning method generally is evaluated on downstream tasks. Thus, the authors should add experiments about this like SpectralNet evaluated on clustering.
>
> **Response:** We focus on the downstream task of visualization, and present experiments on the UMAP extension in Sec. 5.3.
>
> **Comment 7:** In Table 4, the batch size is so large. Thus, I'm concerned about the stability of the model. The authors should analyze this convergence.
>
> **Response:** This approach is standard in graph-based methods (e.g., [6]), as the batches must be representative. It is well-established that this does not affect the scalability or convergence of the method.
>
> **Comment 8:** There are some typo errors including missing definitions of the symbols and missing sequence numbers of the equations.
>
> **Response:** Thank you for pointing this out.
>
> [1] D. Pfau et. al. , “Spectral inference networks: Unifying deep and spectral learning,” ICLR, 2019.
>
> [2] I. Gemp et. al, “Eigengame: Pca as a nash equilibrium,” ICLR, 2021.
>
> [3] Z. Deng et. al, “Neuralef: Deconstructing kernels by deep neural networks,” ICML, 2022.
>
> [4] Z. Chen et. al, “Specnet2: Orthogonalization-free spectral embedding by neural networks,” MSML, 2022.
>
> [5] P. Benner and T. Mach, “Locally optimal block preconditioned conjugate gradient method for hierarchical matrices,” PAMM, 2011.
>
> [6] U. Shaham et. al, “Spectralnet: Spectral clustering using deep neural networks,”, ICLR, 2018.

---

### Author Response · Authors · 2024-11-22
**Thanking the reviewers**

Dear Reviewers and ACs,

We were happy to see that the reviewers found our approach **creative** (ywmX), **extensive** (y9N7) and **addressing important drawbacks in current dimensionality reduction and SE methods** (ywmX, y9N7, WUZ1). . Additionally, we appreciate the reviewers' **encouragement in using our newly proposed evaluation method** - Grassmann Score (ywmX, WUZ1)

We have addressed the reviewer's concerns below.

Thank you

---

### Meta-Review · Area_Chair_g65S · 2024-12-18

**Metareview:**

While appreciating the contribution of the paper, the majority of the reviewers assert that the cons of the paper outweigh its pros. In particular, Reviewer WUZ1 raises serious concerns regarding the motivation validity, experimental validation and technical clarify, and, what is more, the authors fail to address fully the reviewer’s concerns. Hence, I would recommend rejecting the paper.

**Additional Comments On Reviewer Discussion:**

The primary concerns from Reviewer WUZ1 are not addressed properly. For more details, please refer to the author-reviewer discussion form.

---

### Decision · Program_Chairs · 2025-01-22

Reject